# Efficacy of the therapeutic use of video games on the depressive state of stroke patients: Protocol for systematic review and meta-analysis

**Patricia Blázquez-González**[1,2], **Rubén Mirón-González**[3,4], **Alejandro Lendínez-Mesa**[5], **María Nieves Moro-Tejedor**[1,6], **José Luis Cobo-Sánchez**[7,8], **Noelia Mancebo-Salas**[9], **María Teresa Camacho-Arroyo**[5], **Leyre Rodríguez-Leal**[1], **Raquel Luengo-González**[3,4,6]*

1 Red Cross University College of Nursing, Spanish Red Cross, Autonomous University of Madrid, Madrid, Spain, 2 PhD student in Biomechanics and Bioengineering Applied to Health, Doctoral School, University of Alcalá, Madrid, Spain, 3 Department of Nursing and Physiotherapy, University of Alcalá, Madrid, Spain, 4 Group for Research in Community Care and Social Determinants of Health, University of Alcalá, Madrid, Spain, 5 Department of Nursing, University Alfonso X el Sabio, Madrid, Spain, 6 Nursing Research Support Unit, Gregorio Marañón General University Hospital, Madrid, Spain, 7 Nursing Department, Catholic University Santa Teresa de Jesús de Ávila, Ávila, Spain, 8 Development and Innovation Department, Nursing Quality, Training, Research, University Hospital Marqués de Valdecilla, Santander, Spain, 9 Department of Family, Youth and Social Policy, General Office of Social Affairs, Community of Madrid, Madrid, Spain

* raquel.luengo@uah.es

**Data Availability Statement:** No datasets were generated or analysed during the current study. All

## Abstract

### Aim

To assess the effects of virtual reality (VR) on the depressive state of patients with stroke admitted to neuro-rehabilitation units. *Design*: Systematic review and meta-analysis protocol.

### Methods

Randomized Controlled Trials (RCTs) focusing on the effects of virtual reality on depressive state as a primary outcome will be included. Grey literature and the following databases will be consulted: PubMed, Cinahl, PsycInfo, Scopus, Embase, Cochrane Library and Web of Science. The recently revised Cochrane risk of bias tool will be used to assess the quality of included studies. Data will be extracted and meta-analyses will be performed within the specific condition of the emotional state of stroke patients admitted to neurorehabilitation units. Meta-regression and subgroup analyses will be used to identify effective modes and patterns of therapy delivery. The approach of assessment, development and evaluation of recommendations will be applied to reach a convincing conclusion.

### Discussion

An accurate, transparent and standardized review process is expected to provide recommendations on the use of VR technology in the healthcare of stroke patients.

relevant data from this study will be made available upon study completion.

**Funding:** The funders had and will not have a role in study design, data collection and analysis, decision to publish, or preparation of the manuscript.

**Competing interests:** The authors have declared that no competing interests exist.

## Impact

Emotional difficulties are common after stroke and have an impact on rehabilitation outcome. VR seems to have an important role in the treatment and depression in neurological patients, as it is able to improve levels of well-being, coping strategies and social relationships. The systematic review may contribute to a more convincing and specific conclusion compared to existing studies of this type.

## Trial registration

*Systematic review registration*: CRD42022303968.

## Introduction

Stroke is the second leading cause of death worldwide; it is the first leading cause of disability and the second leading cause of dementia. The high morbidity and mortality rates associated with stroke have led to the increasing creation of specialised units for its treatment [1]. Around 30% of people who survive stroke require assistance in activities of daily life (ADLs), 20% need help with ambulation, and 16% need institutional care [2]. An early multidisciplinary approach reduces high dependency rates [3].

The World Health Organization (WHO) defines depression as a condition that can often become chronic or recurrent. Those who suffer depression may feel a lack of concentration, tiredness, guilt, a lack of self-esteem and concentration, sleep or appetite disorders, a lack of interest or pleasure, sadness, hindering the individual's ability to cope with daily life, and in its most severe form, it may even lead to suicide [4].

According to the latest published literature reviews, stroke patients have a prevalence of 19.5% of minor depression and 21.7% of major depression. Furthermore, the loss of autonomy is the most strongly correlated variable in these emotional disorders. [5]. Depression after a stroke is a serious complication that significantly restricts rehabilitation outcomes [6], being the most common neuropsychiatric consequence, with an incidence of approximately 30%–60% after the event [7].

The diagnosis of depression in neuro-rehabilitation units is difficult to manage given the characteristics of the stroke patient. These patients exhibit altered cognitive status, aphasia, agnosia, apraxia and memory disorders. Some of the signs prevalent in depression and stroke are common, such as sleep disturbances, difficulty in concentration, and appetite reduction; others make its aetiology indistinguishable, with it being unclear whether they are consequences of depression, stroke or age. Given these difficulties in diagnosis, between 50% and 80% of cases go undiagnosed [8].

Currently we can find evidence on some digital technologies that can be applied to support traditional care for various neurological dysfunctions showing benefits, not only in sensorimotor and cognitive functions, but also in other problems as pain or depression. But, when it comes to the use of Virtual Reality (VR), there is still no consensus on the types, duration, and intensity of training to assess clinical efficacy [9]. Furthermore, the use of digital technology in stroke patients has been reinforced by the Covid-19 pandemic as an opportunity to accelerate the process of digitalization in the stroke field [10].

Specifically, it has been shown that the use of Virtual Reality (VR) as a co-adjuvant therapy of neuro-rehabilitation in stroke patients with emotional disorders decreases the incidence of

these disorders [11]. VR is considered a cost-effective tool for the treatment of psychological disorders. The reviewed studies claim that VR has positive effects on the affective state of patients. Emotional changes may occur gradually due to the properties and characteristics of the game (immediate feedback, challenging tasks, revaluation, etc.) and may motivate the rehabilitation process [12,13].

The reviewed studies also claim that VR as an adjunct treatment has positive effects, but no evidence supports the use of VR as a sole alternative treatment for mood disorder, anxiety, and depression [14,15].

## Primary objective

This systematic review attempts to assess the effects of VR on the depressive state of patients with stroke admitted to neuro-rehabilitation units by synthesizing the RCTs available.

## Review question

**Primary review question.**   Is the use of VR effective in the treatment of the depression state of stroke patients compared to patients who do not have this adjuvant therapy in their rehabilitation treatment?

**Secondary review question.**   Do the characteristics of the intervention session (number of sessions, duration of the session, individual versus group session) have an impact on your emotional state?

## Methods

We plan to perform this systematic review to the procedures in the Cochrane Handbook.

We also plan to use the MeaSurement Tool to Assess systematic Reviews 2 [16], Preferred Reporting Items for Systematic reviews and Meta-Analyses [17], PRISMA-P (for protocols) 2015 [18]. The PRISMA-P checklist is shown in the S1 Appendix. This protocol has also been registered at the International Prospective Register of Systematic Reviews (PROSPERO) with the registration number as CRD42022303968. The systematic review has been funded since January 2022 and this protocol was reviewed by OML, as external expert in the field.

### Data sources and search strategy

Studies will be identified through electronic searches of bibliographic databases and reference lists of articles with systematically searched in PubMed, Cinahl, PsycInfo, Embase, Cochrane Library and Web of Science will be used.

Medical subject heading (MeSH) terms related and text words will be adopted, mainly including "stroke" OR "brain infraction" AND "virtual reality" OR "virtual reality therapy". No language restriction will be applied. Grey literature such as theses and articles located via the snowball, dissertations and conference proceeding method will be consulted. Experts will be consulted OML on the subject and RLG on the methodological level.

The search will be restricted to include documents published until January 1sr, 2022. The search strategy used for the PubMed database is shown in Table 1. The search strategy for all the databases is shown in the S2 Appendix.

### Criteria for study selection

For the selection of articles there will be no language restriction.

In general, studies will be screened and selected based on PICOS format as follows.

**Table 1. Search methods.**

| Database | Descriptors | Filters |
|---|---|---|
| PubMed | (((((((((((((((("Stroke"[Mesh]) OR "Hemorrhagic Stroke"[Mesh]) OR "Embolic Stroke"[Mesh]) OR "Ischemic Stroke"[Mesh]) OR ("Stroke, Lacunar"[Mesh] OR "Thrombotic Stroke"[Mesh])) OR "Infarction, Posterior Cerebral Artery"[Mesh]) OR "Brain Stem Infarctions"[Mesh]) OR "Infarction, Middle Cerebral Artery"[Mesh]) OR "Infarction, Anterior Cerebral Artery"[Mesh]) AND "Virtual Reality"[Mesh]) OR "Virtual Reality Exposure Therapy"[Mesh]) OR "Exergaming"[Mesh]) | Clinical Trial |

## Types of studies

- Only studies with the design of randomized and non-randomized controlled trials with at least two groups in which one of them evaluates the usual intervention plus the use of virtual reality and a control group with the usual intervention will be included.

- Studies evaluating any degree of intensity and different duration of virtual reality training will be included.

## Types of participants

Adults (at least 18 years of age) of any gender patients admitted to neuro-rehabilitation units due to ischemic or haemorrhagic stroke, all levels of severity, and at all stages after stroke.

## Types of interventions

- We will include studies using virtual reality interventions that met the following definition ""an advanced form of human-computer interface that allows the user to 'interact' and 'immerse' in a computer-generated environment in a natural way" [19].

- Studies using any form of immersive or non-immersive virtual reality, and studies using commercially available game consoles will be included.

- The included studies will measure patients' depression before and after the use of virtual reality.

- Control interventions. Conventional rehabilitation treatment without the use of VR. Given the wide range of alternative interventions, activities that include any activity designed to be therapeutic at the level of impairment, activity or participation that did not include the use of virtual reality will be considered.

## Types of outcome measures

Primary outcomes: in the mental state, we can distinguish three main components: level of consciousness and alertness, cognitive function, and affective state or mood [20]. Accordingly, only articles that focus on the emotional state, specifically depression, will be included in this study. Studies which consider other indicators as a primary outcome will be excluded.

## Screening procedures of eligible studies

After the initial systematic search records will managed with bibliographic manager Mendeley, duplicates will be identified and deleted. All titles identified with the literature search will be included. A peer review (NMS and TCA) will be screened and in case of disagreement, it will

be assessed by a third reviewer (LRL). A justified list of those articles evaluated in full but excluded from the review will be provided. It will be presented in sufficient detail to complete a PRISMA flow chart and diagram as well as a characteristic table of excluded studies [21]. The Kappa coefficients to measure inter-rater agreement will be both calculated for the processes of titles/abstract selection and full-text screening. Specifically, the extent of between-rater agreements can be judged according to the criteria proposed by Landis and Koch [22]: 0.00–0.20 as 'slight agreement', 0.21–0.40 as 'fair', 0.41–0.60 as 'moderate', 0.61–0.80 as 'substantial', and 0.81–1.00 as 'almost perfect agreement'. The plan of study screening and selection is available in the S1 Fig.

## Data extraction

Data extraction will be performed with a pre-piloted, standardized form.

Data will be independently extracted by two reviewers (RMG and ALM) and will identify studies that meet the inclusion criteria. A third reviewer (PBG) will be resolved cases of author disagreement. We will be contacted when there was missing information. Kappa coefficients will be calculated to measure inter-rater agreement in the title/abstract selection process and to the full-text screening.

The data to be recorded will include study characteristics (type of study, authors, year of publication and journal in which it was published), characteristics of the participants (sample size, sex, mean age, nationality and loss rate), characteristics of the intervention (context in which the intervention was delivered, methodology used to deliver the intervention and costs) and the evaluation of the intervention (which scales were used in the study and at which point in the intervention as well as the data and conclusions of the study).

If after the review of the selected articles, any data essential for the review is not available, the authors will be contacted for a request.

## Assessment of risk of bias

The risk of bias will be assessed through domains defined in the Cochrane Handbook for Systematic Reviews Intervention: there are five domains: domain selection bias (sequence generation, allocation concealment), domain conduct bias (blinding of participants and personnel), domain detection bias (blinding of outcome assessors), domain attrition bias (incomplete outcome data), domain reporting bias (selective reporting of results) and domain other biases [23].

The results will be shown as follows: 'low risk' of bias, 'high risk' of bias, or 'unclear risk' of bias. A 'risk of bias' graph will be included in the Cochrane review, where each of the assessments will be presented [24].

Two reviewers of the research team (RMG and ALM) will independently perform the assessment and a third reviewer (PBG) will be responsible to recheck the discrepancies or disagreement during the assessment process and make a final decision. The Kappa coefficients will also be calculated for the five assessment domains and the overall bias respectively.

## Assessment of heterogeneity

**Data synthesis and analysis.**   According to the extracted data, descriptive statistics will be performed with SPSS 26.0 (IBM Company, Chicago, IL, USA) to explore the characteristics of the included studies, including general information, settings, participants, interventions, controls/comparators, outcomes and follow-ups.

Based on the statistics results, the afore-mentioned information will be summarized in detail. Specifically, for the intervention of VR as an adjunctive treatment for rehabilitation will

be considered in addition to conventional rehabilitation. The main information will be summarized in terms of the duration each session, type of session (individual or non-individual), session frequency, etc. It is expected that the main features of the included RCTs, can be therefore clearly depicted.

We will also attempt to classify the RCTs into groups according, randomized controlled trials (RCTs) and non-randomized clinicals trials (NRCT). To ensure the homogeneity we will perform meta-analyses within each condition group. We attempt to pool the Effect sizes (ES) for functioning. Compressive Meta-Analysis (CMA) will be applied to perform meta-analysis. Standardized mean differences, or ES, will be calculated to explore the effects of VR on the depression state of patients with stroke admitted to neuro-rehabilitation units on functioning. According to Cohen (1988) [22], an ES < 0.2 is defined as negligible effect, $0.2 \leq ES < 0.5$ as small, $0.5 \leq ES < 0.8$ as medium, and $ES \geq 0.8$ as large. For repeated measures, all time point meta-analysis method will be used to pool the follow-up data across the studies at different time points to discern the trend of the effect [25]. Heterogeneity will be assessed through a statistic test, I2 who is the proportion of the variability due to real differences between the estimator's respect to the variability due to chance [26].

The data will be interpreted as follows Cochrane Handbook: 0% to 40% (may not be important), 30% to 60% (may represent moderate heterogeneity), 50% to 90% (may represent significant heterogeneity) and 75% to 100% (considerable heterogeneity) [26].

Thus, if we consider that the studies are sufficiently similar clinically, the random effects model will be applied, otherwise, the SE will be calculated on functioning in each individual study and a narrative summary will be presented.

**Assessment of reporting biases.**   If we find more than ten studies, we will create and examine a funnel plot to explore the possible biases of studies and publications according to the Cochrane Handbook for Systematic Reviews of Interventions. We will use Egger's test [27] to analyse the degree of skewness in the funnel plots.

**Assessment of evidence quality.**   For each pooled or individual ES of functioning, the certainty of the evidence body will be independently rated by two reviewers (LRG and PBG) with GRADE [28]. Any discrepancies will be resolved by a third reviewer (RLG). We will create summary of findings tables with GRADEPro-GDT (https://gradepro.org/)

Information on all primary and secondary outcomes from our review will be included. The quality of the evidence will be assessed using five factors: 1) limitations in trial design and implementation of available trials; 2) indirect evidence; 3) unexplained heterogeneity or inconsistency of results; 4) imprecision of effect estimates; and 5) potential publication bias.

For each outcome the quality of the evidence will be classified according to the following categories: high quality (further research is very unlikely to change our confidence in the effect estimate), moderate quality (additional research is likely to have a significant impact on our confidence in the effect estimate and may change the estimate), low quality (further research is very likely to have a significant impact on our confidence in the effect estimate, and the estimate is likely to change) and very low quality (we are not sure of the estimate). Studies with high or unclear risk of performance and detection bias caused by inadequate blinding will be excluded. The subgroup analysis and investigation of heterogeneity data in the following categories will be presented: age, duration of the intervention and gender. A GRADE evidence profile will be performed at the end of the review to document all the findings for the listed primary and secondary outcomes. The certainty of the evidence from each study will be assessed using the GRADE approach, and one of the following levels will be assigned: high or tall, moderate, low and very low. The evidence will be evaluated in each of these domains as indicated in the GRADE manual. Since the certainty of the evidence is subjective, it will be discussed among the reviewers.

The effect of the intervention will be assessed using relative ratio (RR) and 95% confidence intervals for RCT or Odds Ratio (OR) and 95% confidence intervals for NRCT for the dichotomous variable, while the continuous variable will be merged to mean difference (MD) and 95% confidence intervals.

The data will be adjusted by multiplying the mean values by −1 as some scales increase with disease severity. The coefficient of between-rater Kappa will also be calculated.

## Ethical considerations

This review will not involve private information from individuals and will not affect patient rights therefore does not require ethical approval. The results of this review will be disseminated through peer-reviewed publications and conference reports.

## Validity and reliability

For the design and reporting of this systematic review, we will strictly follow the requirements of Cochrane Handbook [29], AMSTAR2 [16], PRISMA [17], and PRISMA-P [18] to ensure the validity and reliability.

## Discussion

Over the past two decades, the number of studies on the efficacy of VR in mental health treatment has increased. The decrease in cost compared to conventional cognition, emotion, and behavioural treatments has contributed to its inclusion [30].

VR is considered a cost-effective tool for the treatment of psychological disorders and, specifically, the treatment of anxiety. Reviewed studies affirm that VR has positive effects on the affective state of patients and may be a more effective treatment than traditional anxiety treatments. The reviewed studies also claim that VR as an adjunct treatment has positive effects, but no evidence supports the use of VR as a sole alternative treatment for mood disorder, anxiety and depression [15,31–33].

Stroke patients are often isolated in their daily life from their family and society. Depression, apathy and anxiety are prevalent conditions in stroke patients, which negatively affect their rehabilitation. The use of VR in rehabilitation allows the patient to receive immediate feedback on task performance as well as visual and audible stimulation and often arouses the patient's interest. VR generates motivation in treatment participation, and the patients have fun while actively performing tasks. It can provide stability in ADLs, thus improving the autonomy of the stroke patient [15,32–35]. In addition, to improving the functional sphere, VR used in the treatment of depression in stroke patients improves their interpersonal relationships when used in a group setting [33].

The average cost of hospitalization in specialized neurorehabilitation units is 4.348 euros per patient, with an average daily expenditure of 194 euros. Expenditure increases 4 times more in totally dependent patients when compared with more independent patients, so the cost-effectiveness of early rehabilitation after stroke are positively associated with the degree of motor disability [36].

Therefore, VR is shown as a therapeutic approach that brings benefits to the rehabilitation process of stroke patients, not only in terms of sensorimotor or cognitive functionality, but also with a positive impact on the emotional situation of these patients [37]. However, it is necessary to synthesize the evidence regarding its benefits taking into account the different types, duration and intensity of training to ensure clinical efficacy [9].

The COVID-19 emergency has highlighted the need to develop and implement new digital technologies for acute and chronic patient care [10]. This review aims specifically to clarify the scientific evidence on the impact of virtual reality on depression symptomatology.

## Supporting information

**S1 Fig. Plan of study screening and selection process.**
(DOCX)

**S1 Appendix. The PRISMA-P checklist and PRISMA flow diagram of the study selection process.**
(DOCX)

**S2 Appendix. The search strategy.**
(DOCX)

## Acknowledgments

The authors would like to thank Dr. Olga Martínez-López (OML) of the University of Castilla-La Mancha for reviewing this protocol as an expert researcher in the field.

## Author Contributions

**Investigation:** Patricia Blázquez-González, Rubén Mirón-González, Alejandro Lendínez-Mesa.

**Methodology:** María Nieves Moro-Tejedor, Noelia Mancebo-Salas, María Teresa Camacho-Arroyo, Leyre Rodríguez-Leal.

**Project administration:** Patricia Blázquez-González.

**Resources:** José Luis Cobo-Sánchez.

**Supervision:** José Luis Cobo-Sánchez, Raquel Luengo-González.

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
