## [Decision Letter · Decision Letter 0]

2 Aug 2022

PONE-D-22-00761Efficacy of the therapeutic use of video games on the depressive state of stroke patients: protocol for systematic review and meta-analysis protocolPLOS ONE

Thank you for submitting your manuscript to PLOS ONE. After careful consideration, we feel that it has merit but does not fully meet PLOS ONE’s publication criteria as it currently stands. Therefore, we invite you to submit a revised version of the manuscript that addresses the points raised during the review process.

We look forward to receiving your revised manuscript.

Kind regards,

Luigi Lavorgna

Academic Editor

PLOS ONE

Journal Requirements:

Reviewers' comments:

Reviewer's Responses to Questions

**Comments to the Author**

1. Does the manuscript provide a valid rationale for the proposed study, with clearly identified and justified research questions?

Reviewer #1: Yes

2. Is the protocol technically sound and planned in a manner that will lead to a meaningful outcome and allow testing the stated hypotheses?

Reviewer #1: Yes

3. Is the methodology feasible and described in sufficient detail to allow the work to be replicable?

Reviewer #1: Yes

4. Have the authors described where all data underlying the findings will be made available when the study is complete?

Reviewer #1: Yes

5. Is the manuscript presented in an intelligible fashion and written in standard English?

Reviewer #1: No

6. Review Comments to the Author

You may also provide optional suggestions and comments to authors that they might find helpful in planning their study.

Reviewer #1: The article by Gonzalez et al. covers a very relevant topic. Methods are sounds. However, I have some comments:

“The high morbidity and mortality rates associated with have led to the increasing creation of specialised units for its treatment.” The sentence lacks a word. I think “stroke”.

“Studies comparing two different types of virtual reality without an alternative group will be not include”. Inclded not include.

However, a revision by a native speaker is definetly needed. In the present form the article is not suitable for publication.

A recent review extensively discusses the use of digital therapeutics in the management of patients with chronic neurological disorders, with a specific focus on VR (PMID: 34018047). A further recente article summarize the application of telemedicine in stroke patients’ management (PMID: 33433756). They deserve to be mentioned.

In the “data sources and search strategy” section, the verb tenses are not congruent. Please, revised it.

List exclusion criteria and inclusion criteria that so scattered are not immediate.

Conclusions should be better related to the purpose of the review.

7. PLOS authors have the option to publish the peer review history of their article (what does this mean?). If published, this will include your full peer review and any attached files.

Reviewer #1: No

---

## [Author Response · Author response to Decision Letter 0]

15 Sep 2022

Responses to comments can be found in the attached document.

---

## [Editor Report · Decision Letter 1]

22 Sep 2022

Efficacy of the therapeutic use of video games on the depressive state of stroke patients: protocol for systematic review and meta-analysis

PONE-D-22-00761R1

We’re pleased to inform you that your manuscript has been judged scientifically suitable for publication and will be formally accepted for publication once it meets all outstanding technical requirements.

Kind regards,

Luigi Lavorgna

Academic Editor

PLOS ONE
---

## [Editor Report · Acceptance letter]

11 Oct 2022

PONE-D-22-00761R1 

Efficacy of the therapeutic use of video games on the depressive state of stroke patients: protocol for systematic review and meta-analysis 

Dear Dr. Luengo-González:

I'm pleased to inform you that your manuscript has been deemed suitable for publication in PLOS ONE. Congratulations! Your manuscript is now with our production department. 

Kind regards, 

on behalf of

Dr. Luigi Lavorgna 

Academic Editor

PLOS ONE